# The role of cell geometry and cell-cell communication in gradient sensing

**Jonathan Fiorentino**[1,2,3¤], **Antonio Scialdone**[1,2,3]*

**1** Institute of Epigenetics and Stem Cells, Helmholtz Zentrum München; München, Germany, **2** Institute of Functional Epigenetics, Helmholtz Zentrum München; Neuherberg, Germany, **3** Institute of Computational Biology, Helmholtz Zentrum München; Neuherberg, Germany

¤ Current address: Center for Life Nano- & Neuro- Science, Istituto Italiano di Tecnologia; Rome, Italy
* antonio.scialdone@helmholtz-muenchen.de

**Data Availability Statement:** The data and code for reproducing all the results of the paper are available at the Github repo https://github.com/ScialdoneLab/2DLEGI.

**Funding:** JF was supported on a Joachim Herz Stiftung Add-on Fellowship for Interdisciplinary Life

## Abstract

Cells can measure shallow gradients of external signals to initiate and accomplish a migration or a morphogenetic process. Recently, starting from mathematical models like the local-excitation global-inhibition (LEGI) model and with the support of empirical evidence, it has been proposed that cellular communication improves the measurement of an external gradient. However, the mathematical models that have been used have over-simplified geometries (e.g., they are uni-dimensional) or assumptions about cellular communication, which limit the possibility to analyze the gradient sensing ability of more complex cellular systems. Here, we generalize the existing models to study the effects on gradient sensing of cell number, geometry and of long- versus short-range cellular communication in 2D systems representing epithelial tissues. We find that increasing the cell number can be detrimental for gradient sensing when the communication is weak and limited to nearest neighbour cells, while it is beneficial when there is long-range communication. We also find that, with long-range communication, the gradient sensing ability improves for tissues with more disordered geometries; on the other hand, an ordered structure with mostly hexagonal cells is advantageous with nearest neighbour communication. Our results considerably extend the current models of gradient sensing by epithelial tissues, making a step further toward predicting the mechanism of communication and its putative mediator in many biological processes.

## Author summary

Groups of cells collectively migrate in many biological processes, ranging from development to cancer metastasis. The migration is often driven by the gradient of a signaling molecule that can be shallow and noisy, raising the question of how cells can measure it reliably. Cellular communication has recently been suggested to play a key role in gradient sensing, and mathematical models with simplified cellular geometries have been developed to help interpret and design experiments. In this work, we generalize the existing mathematical models to investigate how short- and long-range cellular communication can increase gradient sensing in two-dimensional models of epithelial tissues. We analyze various cellular geometries and tissue size, and we identify the optimal setting that

Science (https://www.joachim-herz-stiftung.de/). This work was funded by the Helmholtz Association (https://www.helmholtz.de) to AS. The funders had no role in study design, data collection and analysis, decision to publish, or preparation of the manuscript.

corresponds to different types of communication. Our findings will help pinpoint the communication mechanisms at work in a given tissue and the properties of the molecules that mediate the communication.

## 1 Introduction

Collective migration of epithelial cells is central in plenty of biological processes, ranging from development, to wound healing and cancer [1–11]. The coherent movement of cells is often accompanied by extensive changes in cell shape and geometry, with the formation of higher-order vertices, as in multicellular rosettes [6, 12–15]. In many cases, cells follow the gradient of a guidance cue (e.g., chemoattractants), which induces the initiation of a migration process [4, 16]. The gradients of the chemoattractant signals are typically shallow [17], which raises the question of how cells can detect small changes in chemical concentrations.

Based on experimental evidence, it has been proposed that cellular communication can enhance the ability of groups of cells to measure shallow gradients [18]. The local-excitation global-inhibition (LEGI) model [17–19] is one of the simplest models of collective gradient sensing through intercellular communication. The LEGI model posits that a local and a global reporter molecules are produced in response to the external signal. The local reporter remains confined in the cell and represents a local measurement of the signal; the global reporter is exchanged between neighbour cells, allowing the system to perform a spatial average of the signal concentration. Each cell can then estimate local deviations of the signal from the overall spatial average by combining the information encoded in the levels of the two reporters [18]. This model has been successfully used to describe branching morphogenesis of the epithelial tissue in mammary glands [18], and its physical limits in terms of the properties of the external signal, the cell size and the typical length scale of cellular communication (which in turn depends on the kinetic rates of the LEGI global reporter), have been derived [19].

The ability to measure gradients can improve even further when the local reporter is also exchanged between neighbour cells, as shown with the regional-excitation global-inhibition (REGI) model [19], or when the communication can involve non-nearest neighbours cells [20].

While the above mentioned studies provided experimental and theoretical foundations for the role of cell communication in gradient sensing, they only considered 1-dimensional chains of cells or very simplified 2-dimensional geometries (e.g., made by multiple 1-dimensional chains, coupled in the direction transverse to the gradient [21]). However, complex and dynamically changing geometries are typically found in 2-dimensional epithelial sheets during tissue morphogenesis and organ formation. For example, during the establishment of the anterior/posterior axis in mouse embryos, changes in cell packing occur concurrently with the migration of the anterior visceral endoderm, a subset of cells that specifies the anterior side of the embryo. This leads to the formation of multi-cellular rosettes, i.e., groups of five or more cells that meet at a central point [6]. An increase in disorder of cell packing and shape is also observed in the germband extension in Drosphila [22], in the branching morphogenesis of the developing kidney [23] and in epithelial tube formation [24]. If and how these extensive rearrangements and the generation of higher-order vertices impact gradient sensing is still unknown.

Moreover, most of the studies make the assumption of nearest neighbour communication, which is usually achieved through juxtacrine signalling mechanisms such as via gap junctions [18, 25]. However, long-range signalling is also widespread in epithelial tissues. In this context, the most studied signalling molecules in epithelia are ATP [26], extracellular calcium ions [27, 28] and nitric oxide [29], which diffuse to target receptors through the extracellular space.

Other examples include signalling through soluble ligands that bind the epidermal growth factor receptor (EGFR) in many tissues and organs [30] and the long-range communication mediated by extracellular vesicles in retinal pigment epithelial cells [31]. Moreover, experimental evidence of long-range, in some cases diffusive, communication associated to several signalling pathways, such as Delta-Notch, Nodal and Wnt, is available at present in several species like Drosophila, Zebrafish and several mammals or mammalian cell cultures. The diffusion coefficients of the Notch ligand Delta-like 1 (*Dll1*) have been measured in mammalian cells in culture, both in the plasma membrane and in filopodia [32]. Nodal-Lefty signalling is fundamental for pattern formation in a plenty of organisms; Nodal is a short-range activator and Lefty is a long-range inhibitor. The diffusion properties of Nodal and Lefty have been studied in Zebrafish embryogenesis [33, 34], in chick embryos [35], mice [36, 37] and a synthetic mammalian reaction-diffusion pattern-forming system [38]. All these studies evidence the capability of Lefty proteins to diffuse over long distances, compared to Nodal proteins. Other examples of long-range communication include the extracellular dynamic of Wnt proteins [39], such as *Wnt8* in Xenopus embryos [40–42], which is known to be exchanged over long distances through several mechanisms including diffusion.

Here, we investigate how the size and the geometry of a 2D system of epithelial cells affect the gradient sensing ability. To this aim, we generalize the LEGI model and, via numerical simulations and analytical calculations, we estimate how the signal-to-noise ratio is affected by changes in the mean polygon number, which determines the presence of higher order vertices. In addition to the nearest neighbour communication included in the standard LEGI model, we also consider cellular communication mediated by a global reporter that diffuses in the intercellular space and can thus mediate a long-range communication. Our work integrates in a single modelling framework the complex arrangement of cells in epithelial tissues and different mechanisms of intercellular communication. By doing so, we show how the precision of gradient sensing changes with cell rearrangements and in different regimes of the parameters controlling the communication process, with a potential impact on the understanding of key processes in tissue morphogenesis.

## 2 Materials and methods

### 2.1 Definition of the model

We start from the LEGI model without temporal integration [18, 21] and we consider a molecular signal in a 2-dimensional space with a concentration $c$ that varies with the position $\vec{r}$:

$$c(\vec{r}) = c_0 + \vec{g} \cdot \vec{r} \quad, \tag{1}$$

where $c_0$ is a background concentration and the vector $\vec{g}$ indicates the direction and strength of the gradient.

A group of $N$ epithelial cells respond to this molecular signal by producing a global and a local reporter molecules, in an amount that is proportional to the signal concentration. The global reporter is exchanged between cells, while the local reporter is confined within a cell. Hence, the geometric configuration of the $N$ cells can only influence the dynamics of the global reporter (see Section 2.2).

The stochastic dynamic equations in the linear response regime for the molecule numbers of the local ($u$) and the global ($v$) reporters are [18]:

$$\dot{u}_n = \beta c(\vec{r_n})a^3 - \mu u_n + \eta_n \quad, \tag{2}$$

$$\dot{v}_n = \beta c(\vec{r_n})a^3 - \mu \sum_{n'=1}^{N} M_{nn'}v_{n'} + \xi_n \quad, \tag{3}$$

where $n = 1, \ldots, N$ is the cell index; $c$ is the concentration of the signal; $\vec{r}_n$ is the position of the centroid of cell $n$; $a$ is the linear size of a cell; $\beta$ and $\mu$ are the production and degradation rates of the reporters, respectively; $\eta_n$ and $\xi_n$ are Langevin noise terms [18].

The matrix $\mathbf{M} = \{M_{nn'}\}$ includes terms related to the degradation and the exchange of the global reporter between neighbour cells. Its elements are:

$$
M_{nn'} = \begin{cases} 1 + \sum_{n'=1}^{N} \dfrac{\gamma_{nn'}}{\mu} & \text{if } n = n' \\[2mm] -\dfrac{\gamma_{nn'}}{\mu} & \text{if } n \neq n' \ , \end{cases}
\tag{4}
$$

where $\gamma_{nn'}$ is the exchange rate of the global reporter between cells $n$ and $n'$. In the next sections, we will indicate the matrix of the exchange rates as $\mathbf{\Gamma} = \{\gamma_{nn'}\}$.

In the original unidimensional formulation of the model, the exchange rate is the same for every pair of nearest neighbour cells and the matrix $\mathbf{M}$ is tridiagonal, since it includes the exchange rate of the global reporter between nearest neighbour cells in 1-dimension and its degradation rate. In Section 2.3 we will extend its definition to nearest neighbour communication in more complex 2-dimensional systems and to communication through a reporter diffusing in the intercellular space.

In the LEGI model, the cellular response to the signal is mediated by a molecule that is activated by the local reporter and inhibited by the global reporter. Thus, in the limit of a shallow gradient, the readout in the cell $n$ is the difference between the numbers of the local and global reporter molecules, $\Delta_n = u_n - v_n$ [18]. Given the properties of the reporters, this quantity is equivalent to the difference between the estimation of the local concentration of the signal (provided by the local reporter) and the average concentration (provided by the global reporter) over a spatial region whose extension depends on the range of cell communication [18].

To take into account the noise in the production, degradation and exchange of molecules, the average $\bar{\Delta}_n$ and the variance $(\delta\Delta_n)^2$ of $\Delta_n$ over time are computed from the steady state equations for $u_n$ and $v_n$ [18]:

$$
\bar{\Delta}_n = \bar{u}_n - \bar{v}_n \ ,
\tag{5}
$$

$$
(\delta\Delta_n)^2 = (\delta u_n)^2 + (\delta v_n)^2 - 2\mathrm{cov}(u_n, v_n) \ .
\tag{6}
$$

The quantity of interest is the square root of the Signal to Noise Ratio (SNR) computed in cell $n$

$$
\sqrt{SNR_n} = \left| \frac{\bar{\Delta}_n}{\delta\Delta_n} \right| \ ,
\tag{7}
$$

which quantifies the precision of gradient sensing achieved by cell $n$. To compute it, we use Eqs (5) and (6), following the derivation presented in [18].

The values of the parameters were chosen as in previous studies [18, 19, 43], and are reported in Table 1. We assume the limit of shallow gradient so we take $c_0 \gg a|\vec{g}|$.

**2.1.1 Definition of the model readout.** Given the signal gradient $\vec{g}$, we estimate the SNR in the cell that is exposed to the highest concentration of the signal (as measured in its centroid). This is in line with what has been done in previous studies [18, 19, 21] and presupposes that the cell measuring the highest signal concentration is the first responding (e.g., by starting a migration or branching process).

**Table 1. Parameter values.** $c_0$ is the background concentration, $a$ is the typical linear size of a cell, $\beta$ and $\mu$ are the production and degradation rates of the reporters, $|\vec{g}|$ is the slope of the gradient [18, 19, 43].

| | |
|---|---|
| $c_0$ | $10\ nM$ |
| $a$ | $10\ \mu m$ |
| $\beta$ | $1\ s^{-1}$ |
| $\mu$ | $0.1\ s^{-1}$ |
| $|\vec{g}|$ | $0.04\ nM/\mu m$ |

To account for the different orientations of the gradient, we parametrize it with the angle $\theta$ that the gradient forms with the x-axis (see Fig 1) and sampled 500 equally spaced values of $\theta$ between 0 and $2\pi$. Then, for each value of $\theta$, we compute the SNR as explained above. By doing so, each group of cells has a set of 500 SNR values that represent how the SNR varies over all possible orientations of the gradient.

## 2.2 Generation of 2-dimensional epithelial sheets

We generated cell configurations resembling 2-dimensional epithelial sheets with different number of cells and mean polygon number by using the Python package "tyssue" (https://github.com/DamCB/tyssue, version 0.7.1) [44] as described below.

First, we generated a 2-dimensional configuration with a given number of hexagonal cells and a Gaussian noise on the position of the cell centroids (with zero mean and standard deviation 0.5; "noise" optional parameter in the function "planar_sheet_2d" from the "tyssue" package). By definition, such a configuration has mean polygon number 6. Each cell configuration is defined by the coordinates of the cell centroids, the set of edges forming the boundaries of the cells and the set of vertices in which two or more edges meet. Next, we randomly collapse edges until the target mean polygon number is reached. Using this procedure, we sample $N_C$ configurations with approximately equally spaced values of the mean polygon number in a specified range. As an example, we report in Fig 2A a set of configurations with number of

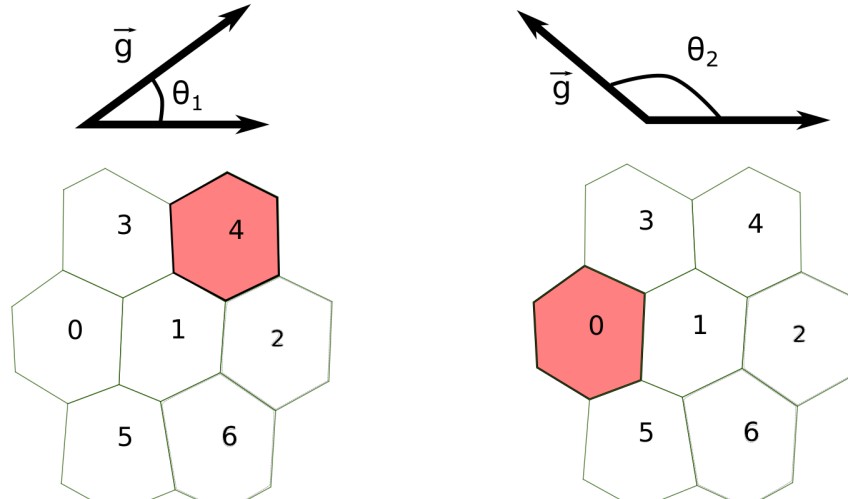

**Fig 1. Computation of the SNR in a 2-dimensional LEGI model.** We account for different orientations of the gradient $\vec{g}$ by computing the SNR for 500 different values of the angle $\theta \in [0, 2\pi]$. For each value of $\theta$, we estimate $\sqrt{SNR}$ in the edge cell, namely the cell measuring the highest concentration of the signal (colored in red). In the examples depicted here, for $\theta = \theta_1$ (left) and $\theta = \theta_2$ (right), the edge cells are cell 4 and cell 0, respectively.

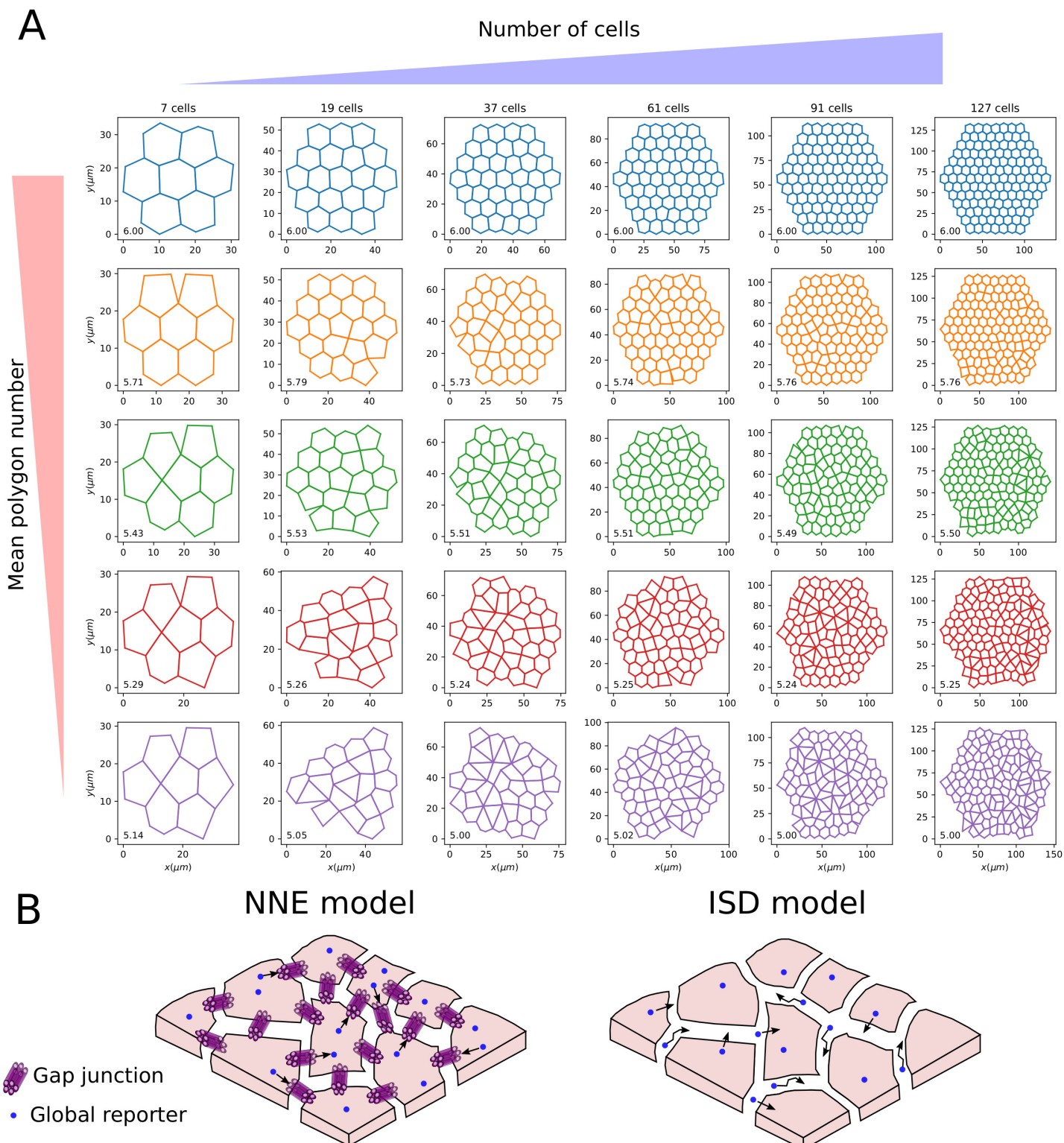

**Fig 2. Generation of 2-dimensional epithelial sheets and intercellular communication modes.** A: Examples of the 2D configurations of cells we generated with increasing cell numbers (from left to right) and decreasing mean polygon number (from top to bottom). The value of the mean polygon number is reported in the bottom left of each configuration. B: Scheme of the communication models studied in the present work: the Nearest Neighbour Exchange (NNE) model (left panel), where a global reporter (blue circle) is exchanged between nearest neighbour cells via, e.g., gap junctions; and an Intercellular Space Diffusion (ISD) model, in which the global reporter is secreted by cells and diffuses in the lateral intercellular space, thus allowing communication between non-nearest neighbours.

cells between 7 and 127 and mean polygon number between 5 and 6. All the configurations used in the present study are available from the GitHub repo https://github.com/ScialdoneLab/2DLEGI.

### 2.3 Two modes of communication

In this Section, we introduce the two models of intercellular communication we analyzed, which determine the form of the matrix **M** introduced in Section 2.1.

**2.3.1 Nearest neighbour exchange.** The first model of communication is the Nearest Neighbour Exchange (NNE), in which the global reporter is exchanged between nearest neighbour cells. This is the same model considered in [18, 19, 21, 43], although they included simpler system geometries. For the previously studied 1-dimensional chain of cells, the matrix **M** is tridiagonal, since each cell has two nearest neighbours [18, 19]. Here **M** has a more complex structure, given that it needs to take into account the different set of nearest neighbours that each cell has in the 2D configurations. With the NNE model, the matrix of exchange rates of the global reporter, **Γ**, can take only two values: 0 for non nearest neighbour cells and $\gamma$, the exchange rate between nearest neighbours that sets the communication strength. An example of the exchange rates resulting from NNE communication in the configuration shown in Fig 3A is displayed in Fig 3B as a weighted graph.

**2.3.2 Intercellular space diffusion.** Alongside the nearest neighbour communication in the NNE model, we also analyze cell communication via a global molecular reporter that is secreted and can diffuse in the lateral intercellular space, enabling long range communication between non-nearest neighbour cells (see Fig 2B). Diffusion in the lateral intercellular space is known to occur in epithelia and has been studied, for example, in cultured renal cells (Madin-Darby canine kidney) [45, 46]. In this model of cell communication that we call Intercellular Space Diffusion (ISD), the exchange rates $\gamma_{nn'}$ between any pair of cells $n$ and $n'$ can be non-zero and can be written as:

$$\gamma_{nn'} = \alpha P_{nn'} \tag{8}$$

where $\alpha$ is the secretion rate of the reporter and $P_{nn'}$ is the probability that a reporter secreted by cell $n$ is internalized by cell $n'$.

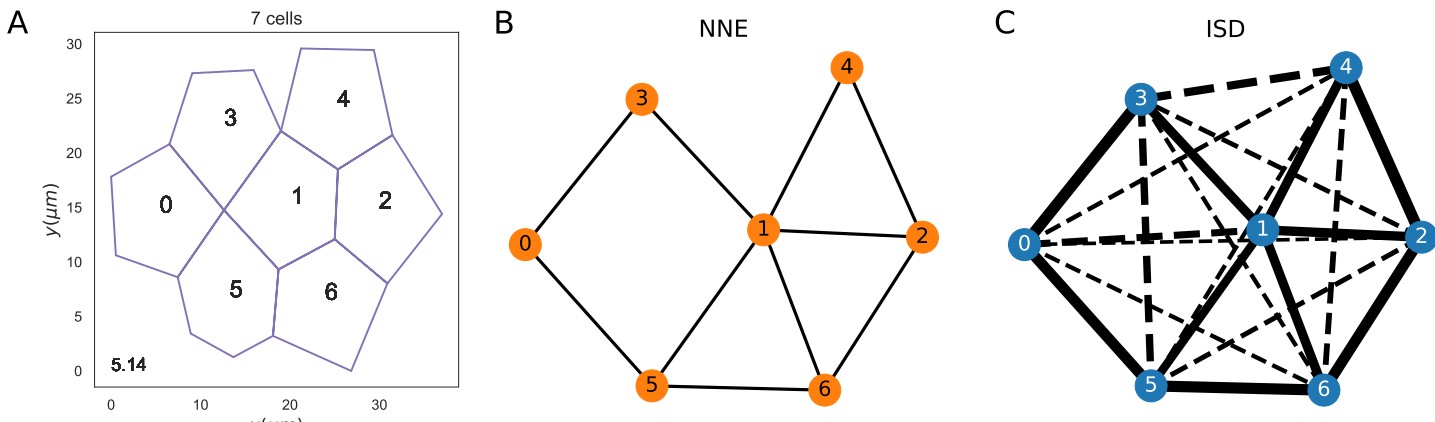

**Fig 3. The exchange rates in two different cell communication models.** A: Configuration of 7 cells with a mean polygon number equal to 5.14. B,C: The exchange rates between the cells of panel A are shown in weighted graphs where nodes are cells and edge thickness represents the magnitude of the exchange rate. Panel B refers to the NNE communication, where only nearest neighbours can exchange the global reporter. Panel C illustrates the exchange rates in the ISD model, where also non-nearest neighbours (joined by dashed edges) can communicate via the global reporter. Parameter values: $\gamma = 0.1 \ s^{-1}$ for the NNE model; $D = 1000.0 \ \mu m^2/s$ and $\alpha = 1.0 \ s^{-1}$ for the ISD model.

The probabilities $P_{nn'}$ depend on the internalization rate $\lambda$, on how fast the reporter can diffuse in the intercellular space and on the geometry of the epithelial tissue (see Fig 3C). To estimate $P_{nn'}$ for a given configuration we resorted to numerical simulations. The simulations include two main events: the diffusion of the reporter in the lateral intercellular space and its internalization by a cell. Below is a schematic description of the simulation steps; the code, along with a more detailed description, is available from the GitHub repo https://github.com/ScialdoneLab/2DLEGI.

1. Sample the internalization time $t_{int}$ from $P_{int}(t) = \lambda e^{-\lambda t}$ (assuming that the reporter internalization is a Poisson process);

2. Consider cell $n$ as the cell releasing the global reporter;

3. Randomly choose an edge from the set of edges $\{e_1, \ldots, e_{N_n}\}$ forming the boundary of cell $n$, with a probability proportional to the edge lengths $\{l_1, \ldots, l_{N_n}\}$;

4. Once a certain edge $e_k$ has been chosen, place the global reporter at a random position along the edge;

5. Perform a 1-dimensional Brownian motion on the chosen edge with diffusion coefficient $D$;

6. If the global reporter reaches a vertex, it can switch edge or stay on the same edge $e_k$, with uniform probability. The set of possible edges is composed of those connected to the vertex that has been reached;

7. When $t = t_{int}$, the cell that internalizes the reporter is sampled, with uniform probability, among the set of cells sharing the edge on which the molecule is diffusing at time $t$;

8. Repeat steps 1.–7. for each cell in the configuration: $n = 1, 2, \ldots, N$.

For each cell in the configuration, we generate $N_T = 5000$ trajectories using the above algorithm, fixing $\lambda = 1.0\ s^{-1}$ and taking $D = 10\ \mu m^2/s$ or $D = 1000\ \mu m^2/s$ to model slow or fast diffusion (see below and [45, 46]).

From the above simulations, we computed the probabilities $P_{nn'}$ as

$$P_{nn'} = N_T^{(nn')}/N_T \quad , \tag{9}$$

where $N_T^{(nn')}$ is the number of simulated trajectories in which the global reporter has been released by cell $n$ and internalized by cell $n'$. These probabilities are then used to compute the exchange rates via Eq (8).

Fig 3C illustrates the values of the exchange rates corresponding to the configuration in Fig 3A in an ISD model, with the dashed lines indicating communication between non-nearest neighbours.

**2.3.3 Statistical analysis.** Once we computed the SNR values associated with the different configurations and communication models, we used the Wilcoxon rank sum test to assess the presence of statistically significant differences. In each statistical test, in addition to the p-value, we computed also the common language effect size (CLES), which is more stable than p-values to variations in the number of sampled configurations [47].

For two sets of observations $A$ and $B$, the CLES is defined as

$$CLES = P(A > B) + 0.5 P(A = B) \quad . \tag{10}$$

We consider the outcome of the Wilcoxon rank sum test significant if the p-value is smaller than 0.05 and the CLES is smaller than 0.4 or larger than 0.6. The Wilcoxon rank sum tests

and CLES are calculated using the Python package "pingouin" [48] (https://pingouin-stats.org/index.html, version 0.3.10).

## 3 Results

### 3.1 Characterization of the ISD communication model

**3.1.1 Analytical estimation of the exchange rates.** While we used computer simulations to evaluate the SNR (see above), here we provide an analytical estimation of the exchange rates of the global reporter in the ISD model, to understand how the parameters affect cellular communication in this model.

Once a molecule is released on a cell boundary (i.e., an edge) in $x = x_0$, under the assumption that the internalization is a Poisson process with rate $\lambda$, we can write the probability that the molecule is internalized at a time $t < \tau$ as:

$$P(0 < t < \tau) = \int_0^\tau \lambda e^{-\lambda t} dt = 1 - e^{-\lambda \tau} \quad , \tag{11}$$

From this, using the First Passage Time Distribution for a free Brownian motion (i.e., a Levy distribution) [49], $FPT(\tau, s)$, we can obtain the probability that the molecule is internalized by a cell before travelling a distance $s$:

$$P_{int}(s) = \int_0^\infty FPT(\tau; s) P(0 < t < \tau) d\tau =$$

$$= 1 - \int_0^\infty \frac{s}{\sqrt{4\pi D \tau^3}} e^{-\frac{s^2}{4D\tau} - \lambda \tau} d\tau = 1 - e^{-\sqrt{\frac{\lambda s^2}{D}}} \quad . \tag{12}$$

Based on $P_{int}(s)$, we can obtain an approximation for $P_{nn'}$, i.e., the probability that a global reporter secreted by cell $n$ is internalized by cell $n'$:

$$P_{nn'} \sim \frac{P_{int}(2l_E)}{N_m} [1 - P_{int}(2(m-1)l_E)] \tag{13}$$

where $n$ and $n'$ are neighbours of order $m$ (i.e., $m = 1$ corresponds to nearest neighbours), $l_E$ is the average edge length and $N_m$ is the number of neighbours of cell $n$ of order $m$. In this approximate calculation, we considered that a molecule typically has to travel at least $\sim 2$ edges to move from a neighbour cell of order $m$ to one of order $m + 1$.

In the present model, diffusion of the LEGI global reporter is restricted to the 1-dimensional lateral intercellular space (LIS). However, this assumption is not as restrictive as it could appear: in S1 Text, we extend the analytical calculations to the case in which global reporter molecules can escape from the LIS. We show that an escape rate $\lesssim 20\%$ of the internalization rate of the molecule has very little effect on the probability of molecular exchanges between cells (see S1 Fig).

**3.1.2 Local and global communication in ISD.** Communication in the NNE model is controlled by the exchange rate of the global reporter $\gamma$, whose ratio with the degradation rate $\mu$ sets the communication strength. In the ISD model, the global reporter has a more complex dynamics: it is secreted with a rate $\alpha$, diffuses in the intercellular space with a diffusion coefficient $D$ and is internalized with a rate $\lambda$. These parameters determine the strength and the range of communication.

In particular, $\alpha$ mainly regulates the strength of communication, since the exchange rates $\gamma$ are proportional to it (see Eq (8)). $D$ and $\lambda$ instead determine how far a global reporter diffuses

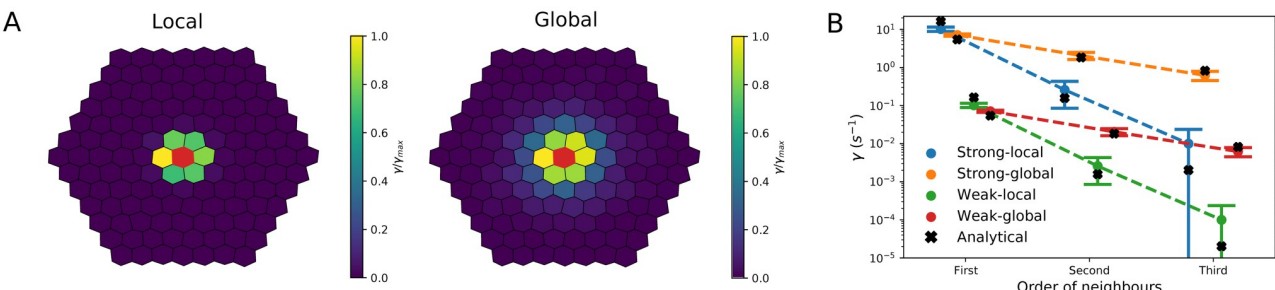

**Fig 4. Communication regimes in the ISD model.** A: We generated a configuration of 127 cells with mean polygon number 6. Here cells are coloured according to the values of their exchange rate $\gamma$ with the central cell (coloured in red), normalized to the maximum $\gamma_{max}$. This is shown in the local ($D = 10\ \mu m^2/s$, left panel) and global ($D = 1000\ \mu m^2/s$, right panel) communication regime. B: Mean and standard deviation of the exchange rates with the central cell of the configuration shown in panel A (y axis), for the n-th neighbours (x axis). Circles represent values obtained from the simulations, while the crosses are the analytical estimations. Four different combinations of parameters are used, corresponding to strong/weak and global/local communication regimes. Parameter values: $\alpha = 1.0\ s^{-1}$ and $\alpha = 100.0\ s^{-1}$ for weak and strong communication; $D = 10\ \mu m^2/s$ and $D = 1000\ \mu m^2/s$ for local and global communication, respectively.

before it is internalized, and hence how far apart two cells can be to communicate, as the analytical estimations of the exchange rates showed (see above section 3.1.1). More specifically, the calculations suggested that the communication is only "local" (i.e., mostly restricted to nearest neighbours) if $D \lesssim l_E^2 \lambda$; conversely, if $D \gtrsim l_E^2 \lambda$ the reporter can reach also cells beyond nearest neighbours and a "global" communication occurs (Eq (13)). An example of the local versus global communication regimes is shown in Fig 4A, where the normalized exchange rates $\gamma$ estimated by simulations are plotted. Fig 4B shows the values of the exchange rates as predicted by the simulations and by analytical calculations (Eq (13)) in the four regimes of the ISD model, characterized by strong/weak and local/global communication.

## 3.2 The precision of gradient sensing in configurations with different cell numbers

We compared the values of $\sqrt{SNR}$ (see section 2.1.1 and Fig 1) for configurations with increasing number of cells (in the range $7 \leq N \leq 217$) and mean polygon number equal to 6, in the NNE and the ISD models. The analysis was carried out on 10 independent sets of random configurations to ensure the robustness of the results.

Roughly speaking, in a LEGI model, the SNR increases with the number of cells $N$ as long as cells on opposite edges can communicate with each other. The reason for this is that a larger system of (communicating) cells can sample more values of concentration over larger areas (see section 2.1). This is the case in the "strong" communication regime, both in the NNE and the ISD model, where the $\sqrt{SNR}$ increases approximately in a linear fashion over the range of $N$ we explored (Fig 5C and 5D and S2A Fig). However, the situation changes in a regime of "weak" communication: in this case, if the communication is "local", i.e., it involves only the nearest neighbour cells, then $\sqrt{SNR}$ reaches a maximum around $N \sim 60$ and then it declines, in both the NNE and the ISD models (Fig 5A and 5B). This result is in line with what Smith et al observed in their 2D model with a simplified geometry [21]. Indeed, with a weak-local communication, adding more cells does not improve much the signal once the system size is above a certain value (see S3A and S3B Fig), while the fluctuations keep increasing (see S3C and S3D Fig), causing an overall decrease in the SNR.

In the weak-global communication regime, which is present only in the ISD model, an intermediate situation is found: even if the communication is weak, it also involves cells

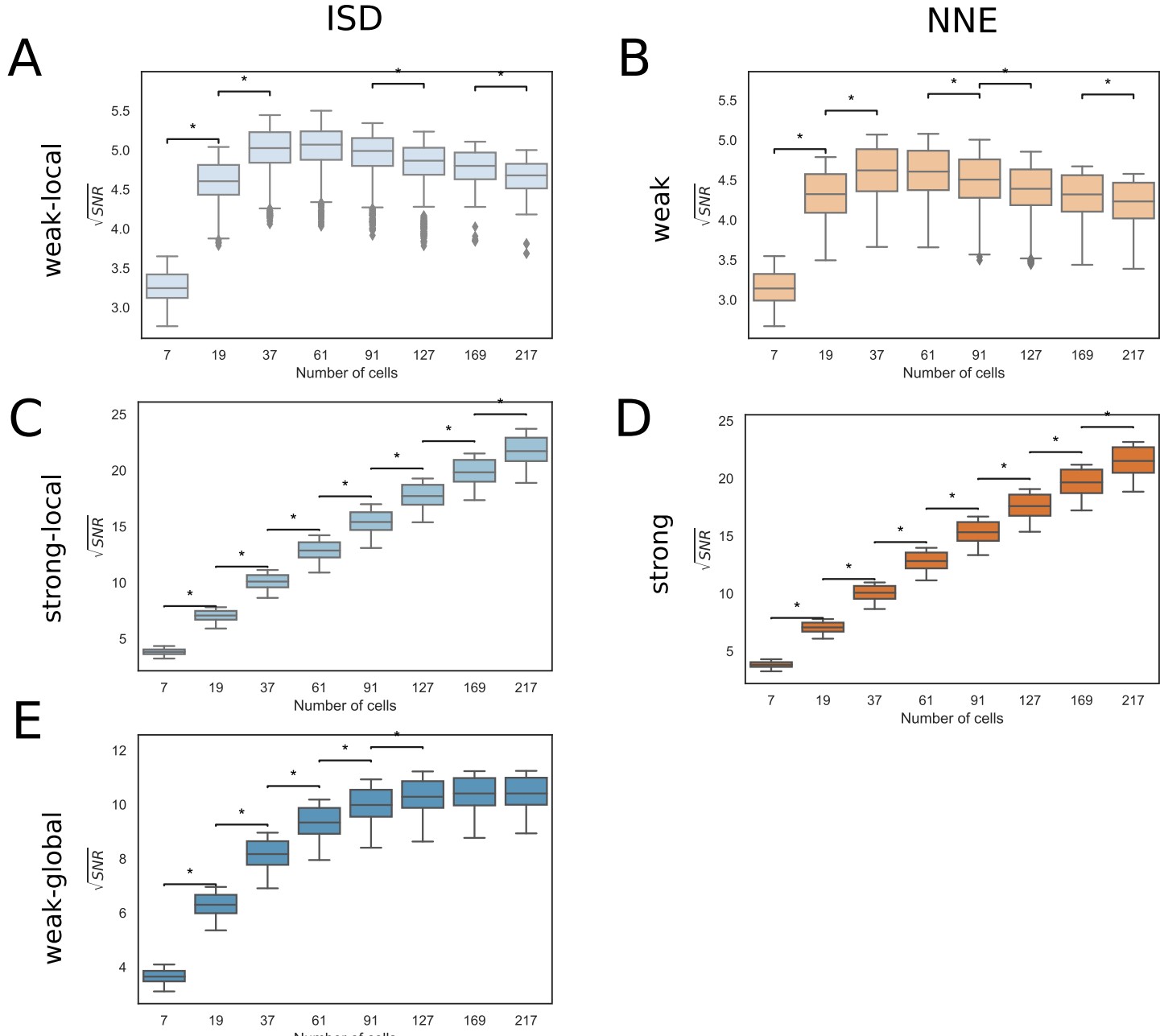

**Fig 5. Dependence of the SNR on the number of cells.** Box plots comparing the set of $\sqrt{SNR}$ values for configurations with different number of cells and mean polygon number 6. The results are obtained by combining up to 10 sets of cell configurations each. A: Weak-local (ISD) communication: $D = 10.0\ \mu m^2/s$, $\alpha = 1.0\ s^{-1}$. B: Weak (NNE) communication: $\gamma_{NNE} = 0.1\ s^{-1}$. C: Strong-local (ISD) communication: $D = 10.0\ \mu m^2/s$, $\alpha = 100.0\ s^{-1}$. D: Strong (NNE) communication: $\gamma_{NNE} = 10.0\ s^{-1}$. E: Weak-global (ISD) communication: $D = 1000.0\ \mu m^2/s$, $\alpha = 1.0\ s^{-1}$. The ⋆ indicates statistically significant results (see Materials and methods for further details), the CLES are reported in S1 Table.

beyond nearest neighbours. Thus, as the number of cells increases, the SNR keeps on increasing even in larger systems (e.g., $60 \lesssim N \leq 127$, see Fig 5E), but it saturates for $N \gtrsim 127$.

These observations are robust to changes to the system geometry, as the trends of SNR as a function of the cell number are the same with different values of the mean polygon number

(S4 Fig). Minimum and maximum values of $\sqrt{SNR}$ corresponding to panels of Fig 5, S2 and S4 Figs are reported in S2 Table.

### 3.3 Differential impact of the mean polygon number on the precision of gradient sensing

Epithelial sheets can have different geometries characterized by specific cell shapes that influence the patterns of cell-cell contacts. The mean polygon number quantifies the number of neighbours (or sides) of a cell, and it typically varies between 5 and 6, as evidenced, for instance, in the Drosophila wing disc [50–52] and during the migration of the anterior visceral endoderm cells in mouse embryos [6].

In this section, we compute how gradient sensing is affected by cell shape looking at the SNR as a function of the mean polygon number in a system with a fixed size ($N$ = 127 cells).

In the strong-local and strong-global communication regimes (Fig 6C and 6D and S2B Fig), the mean polygon number has little effect on the SNR in both the ISD and the NNE models, due to the very efficient exchange of the global reporter between neighbour cells. On the other hand, in the case of weak-local communication, the SNR tends to slowly increase with the mean polygon number, especially in the NNE model (Fig 6A and 6B). This is because, with larger mean polygon numbers, the number of nearest neighbours increases, which is advantageous when the communication is local and not very efficient.

Interestingly, in the weak-global regime of the ISD model, we found the opposite behaviour: the SNR is higher in configurations having lower mean polygon numbers (Fig 6E). This implies that, in this regime, configurations with higher-order vertices, characterized by the presence of heterogeneous cell shapes and of, e.g., multicellular rosettes [6], confer a better ability to sense shallow gradients with respect to more ordered configurations. Indeed, with more disordered configurations and higher-order vertices, the global reporter can reach more efficiently cells that are located at larger distances. To show this, we estimated from our simulations the exchange rates $\gamma$ as function of the distance $d$ between cells in the weak-global regime of the ISD model, and we found that $\gamma(d)$ has a slower decay with a smaller mean polygon number (Fig 6F), which suggests a more efficient long-range communication. This effect could also be partly due to a reduction in the perimeters of cells with smaller polygon numbers (S5 Fig).

The results in systems with fewer cells are consistent with those shown above, although the trends are noisier and less clear (S6 Fig). Minimum and maximum values of $\sqrt{SNR}$ corresponding to panels of Fig 6, S2 and S6 Figs are reported in S4 Table.

### 3.4 Comparison between the NNE and ISD models

Finally, we compared the gradient sensing performance of the NNE and ISD models on the same cell configuration.

To make this comparison, for each of the four regimes of the ISD model (with the parameters selected above; see Fig 5), we computed the average exchange rate between nearest neighbours $\gamma_{ISD}^{nn}$ and in the NNE we fixed the exchange rate $\gamma_{NNE} = \gamma_{ISD}^{nn}$. The range of values obtained for $\gamma_{NNE}$ in each communication regime of the ISD model is shown in Table 2. By doing this, in each comparison we imposed that the global reporter is exchanged at roughly the same rate between nearest neighbours in the two models.

The results with $N$ = 127 cells and different values of the mean polygon number are shown in Fig 7, where on the $x$-axis we plotted the $\sqrt{SNR}$ for ISD and on the $y$-axis the $\sqrt{SNR}$ obtained with NNE.

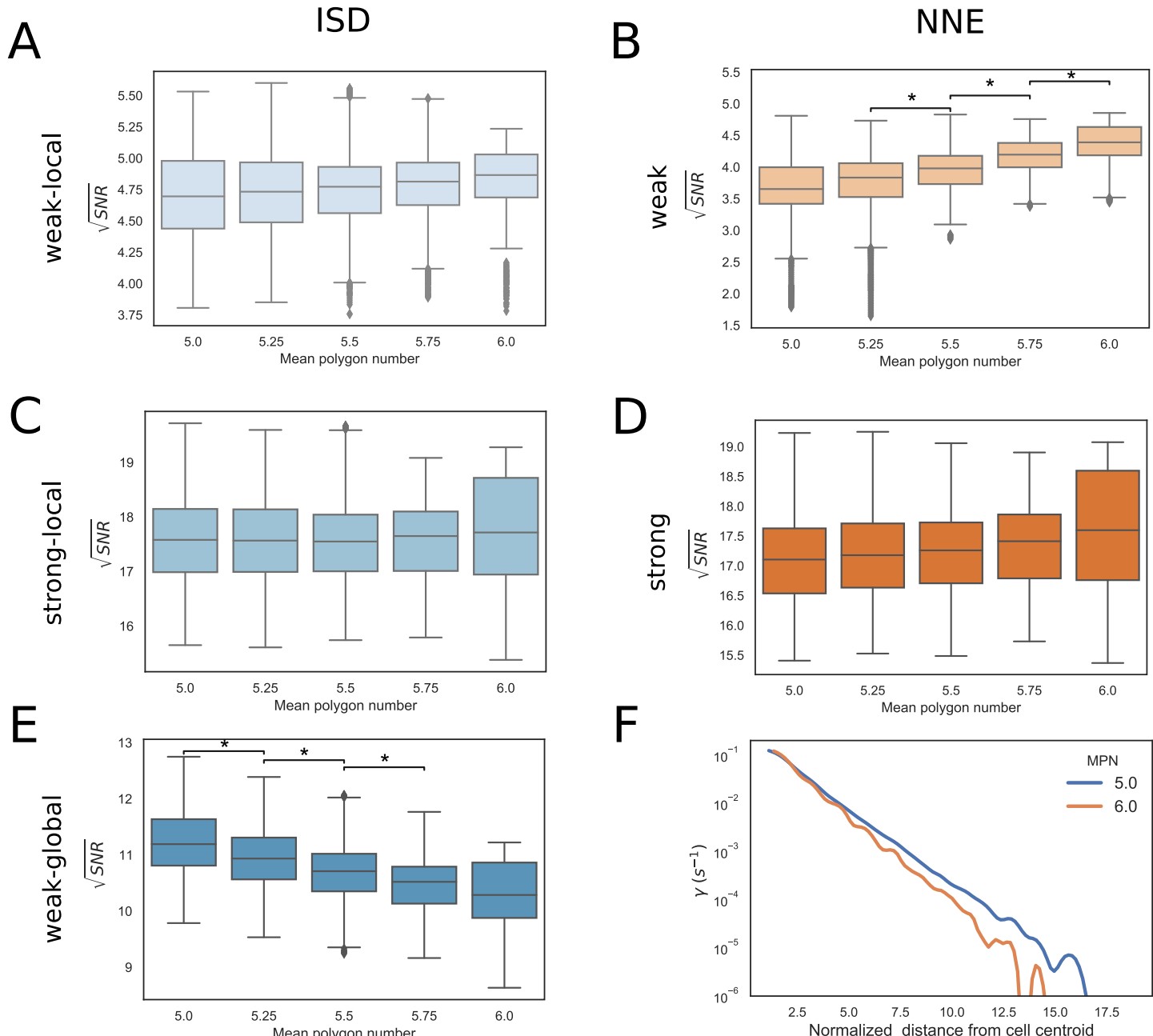

**Fig 6. Dependence of the SNR on the mean polygon number.** Box plots comparing $\sqrt{SNR}$ for configurations with different mean polygon number and 127 cells. The results are obtained combining 10 sets of cell configurations. A: Weak-local (ISD) communication: $D = 10.0 \ \mu m^2/s$, $\alpha = 1.0 \ s^{-1}$. B: Weak (NNE) communication: $\gamma_{NNE} = 0.1 \ s^{-1}$. C: Strong-local (ISD) communication: $D = 10.0 \ \mu m^2/s$, $\alpha = 100.0 \ s^{-1}$. D: Strong (NNE) communication: $\gamma_{NNE} = 10.0 \ s^{-1}$. E: Weak-global (ISD) communication: $D = 1000.0 \ \mu m^2/s$, $\alpha = 1.0 \ s^{-1}$. The $\star$ indicates statistically significant results (see Materials and methods for further details), the CLES are reported in S3 Table. F: Spline fit of the values of the exchange rates in the weak-global regime of the ISD model as a function of the normalized distance from the selected cell centroid. We considered the edge cells of configurations with 127 cells and mean polygon number (MPN) 5 and 6. Distances are normalized by the average edge length $l_E = 7.1 \ \mu m$ in the considered configurations, so that the x-coordinate indicates approximately the order of neighbouring cells.

In general, the SNR tends to be larger in the ISD compared to the NNE case, due to the possibility of a long-range communication in the ISD model. More specifically, when the reporter is mostly only locally and very efficiently exchanged (strong-local regime, Fig 7B), the two models are almost equivalent. ISD yields a greater SNR if a longer range communication is

**Table 2. Exchange rates in the NNE model for the comparison with the ISD model.** Range of values of $\gamma_{NNE}$ employed for the comparisons of communication models in each regime of the ISD model. We considered 10 sets of configurations with 127 cells and different values of the mean polygon number (data shown in Fig 7).

| Communication regime | $\gamma_{NNE}$ $(s^{-1})$ |
|---|---|
| Weak-local | [0.10,0.11] |
| Strong-local | [10.4,11.2] |
| Weak-global | [0.078,0.082] |
| Strong-global | [7.75,8.24] |

enabled (strong-global regime, Fig 7D), or when the communication is less efficient (weak-local and weak-global, Fig 7A and 7C). In particular, the largest difference between the two models is observed in the weak-global regime (Fig 7C): here the exchange rates with the nearest neighbours are $\lesssim 10^{-1}\ s^{-1}$, so the communication in the ISD model profits from the extent of the diffusion of the LEGI global reporter, while the communication in the NNE model is very inefficient. Notably, the advantage of the ISD over the NNE model is larger for smaller mean polygon number, in the weak-local and weak-global communication regimes.

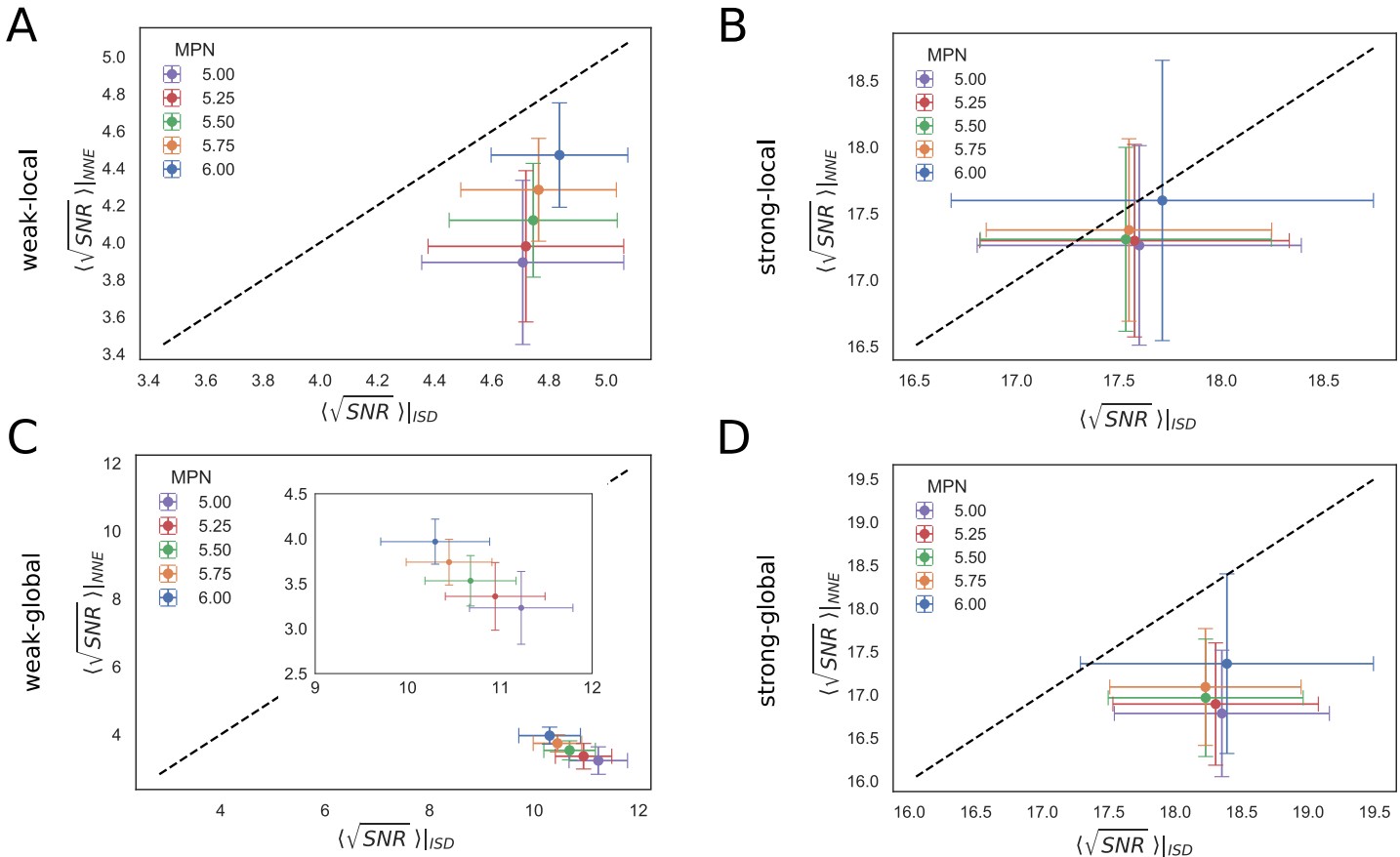

**Fig 7. Comparison between ISD and NNE communication models.** Scatter plots of the mean of $\sqrt{SNR}$ for 10 sets of configurations, for the ISD ($x$-axis) and NNE ($y$-axis) communication models. Results are shown for configurations with 127 cells and different values of the mean polygon number (MPN). The dashed black line marks the bisector and error bars indicate the standard deviation. A: Weak-local communication: $D = 10.0\ \mu m^2/s$, $\alpha = 1.0\ s^{-1}$. B: Strong-local communication: $D = 10.0\ \mu m^2/s$, $\alpha = 100.0\ s^{-1}$. C: Weak-global communication: $D = 1000.0\ \mu m^2/s$, $\alpha = 1.0\ s^{-1}$. The inset zooms in on the $\sqrt{SNR}$ values, showing that the advantage of ISD is greater for smaller MPN. D: Strong-global communication: $D = 1000.0\ \mu m^2/s$, $\alpha = 100.0\ s^{-1}$.

The results for configurations with a different number of cell are shown in S7 Fig. For small configurations the models are more similar, while the advantage of the ISD model becomes more apparent as the number of cells increases, especially in the weak-global regime.

## 4 Discussion

In this paper, we have analyzed the ability to measure shallow gradients of signals by 2D epithelial tissues via cellular communication. We started from a previously proposed model called LEGI, and extended it to study complex cell geometries in 2D and by adding an alternative mechanism enabling long-range communication, based on the diffusion of the global reporter in the intercellular space. Then, we computed the SNR of different cell configurations varying the model parameters and the type of communication.

We found that, with a number of cells $7 < N < 217$, different behaviours of the SNR are possible depending on the type of communication. Specifically, smaller systems tend to have higher SNR when the communication is "weak" and "local", whereas larger systems are better when the communication is "global" (with the ISD communication) and/or "strong" (see Fig 5).

With our model, we generated different cell geometries that are characterized by higher order vertices associated with, e.g., multi-cellular rosettes. This allowed us to characterize how the geometry influences the SNR with different types of communication. Interestingly, we observed that, in the ISD model, when the reporter is exchanged with low efficiency but can diffuse fast in the intercellular space, more irregular geometries have a higher SNR and, thus, are more sensitive to external gradients. This result might explain the observed extensive cell re-arrangements and the formation of multi-cellular rosettes occurring in the anterior visceral endoderm (AVE) of the early mouse embryo concomitantly with its migration, which is essential for the definition of the anterior/posterior axis [6]. Indeed, the potential role of cellular communication in AVE migration has been recently hypothesized [53].

The simplest version of the ISD model assumes that the diffusion of the global reporter is confined in the LIS. There is specific evidence of processes that could confine the diffusion of molecules in the lateral intercellular space. For example, molecules like Wnt proteins can dynamically associate/dissociate from cell membranes and diffuse on them [54], and tight junctions can significantly reduce the outflux of molecules with a size $\gtrsim 3.5$ angstrom [55] (as a reference, the size of ATP is $\sim 14$ angstrom and Wnt proteins $\sim 60–70$ angstrom). However, the ISD model does not require a total confinement of the global reporter in the LIS, and a modest "escape rate" from the LIS has little effects on the long-range communication (see S1 Text and S1 Fig).

Our modelling framework allows to probe different scenarios of the gradient sensing process when experimental measurements of the diffusion coefficients of candidate molecular reporters are available [32–38, 40], or to identify the communication regime in which the biological system under consideration operates, if the rates of secretion and internalization, which concur in controlling the reporter dynamics, are also known.

Several extensions of our model could provide further insights into the biological processes involving gradient sensing, cellular communication and migration. A first step would consist in the combination of our work with more complex biochemical models of cellular communication, as the one proposed for modelling EGFR signalling in Drosophila oogenesis [56].

Moreover, other ways of introducing long-range cell-cell communication are possible: examples include the exchange of molecules through cellular protrusions, such as filopodia or cytonemes [57, 58], or other cellular channels, such as epithelial bridges [59] and tunnelling nanotubes [60, 61]. Our model could account for this means of intercellular communication, replacing the diffusion in the lateral intercellular space with that through cellular protrusions.

In this case, probing different cell configurations varying the mean polygon number would amount to consider different distributions of the number of connections of a cell with neighbours of different order. Finally, mechanical ways of obtaining long-range communication are possible, as observed in the Drosophila ovary with the presence of mechanical feedback through cadherins [62].

Another interesting direction is to consider both cellular communication and migration in the same model. To this end, a Cellular Potts Model on a 2D lattice relying on the original LEGI model with nearest neighbour communication has been proposed [43], while a recent work focused on the role of cell-cell adhesion in the migration of 2D groups of cells [63]. Our generalization of the LEGI model with complex 2D geometries would improve the above mentioned cell motility models. Several other models of collective cell migration including gradient sensing, not relying on the LEGI paradigm, have been developed. For instance, Roy et al [64] have analyzed the influence of connectedness and size of 2D cell clusters on the velocity of migration, and Camley et al [65] studied the role of cell-cell variation in responding to an external signal in collective cell migration. All these models study specific properties of groups of cells that would be worth considering in an extension of our model to include a cell motility mechanism.

In conclusion, here we provide a modelling framework that clarifies the role of the size and the geometry of epithelial cells in gradient sensing, and can be easily extended to include further molecular and geometrical details. We believe it will be particularly interesting to combine this model with the large-scale sequencing and imaging data that are becoming available, for example on AVE migration in mouse embryos [66–68], with the goal of identifying the underlying molecular mechanisms. Many algorithms, in fact, have been developed to extract from single-cell and spatial transcriptomic datasets candidate molecules that could be mediating cellular communication (e.g., [69–71]). Our model can complement these algorithms by providing a quantitative framework that can be used to further refine the lists of candidate molecules based on their properties, the size of the system and the cell geometry. To facilitate the use of our modelling framework by the community, we made all the code freely available on GitHub (https://github.com/ScialdoneLab/2DLEGI).

## Supporting information

**S1 Text. Including global reporter escape in the ISD model.**
(PDF)

**S1 Fig. Cellular communication in the ISD model with a global reporter escaping LIS.**
Probability of reporter internalization before travelling a distance $s$ $\tilde{P}_{int}(s)$ (left), and probability of exchange of the reporter between a pair of neighbouring cells of order $m$, $P_{nn'}$, for different values of the escape rate $v$ and the two values of the reporter diffusion coefficients. $N_m$ is the number of neighbours of the cell releasing the global reporter. A: $N_m = 6$. B: $N_m = 5$. Other parameters are indicated at the top of each panel. See S1 Text for the mathematical details.
(PDF)

**S2 Fig. Results in the strong-global communication regime of the ISD model.** A: Box plots comparing the set of $\sqrt{SNR}$ values for configurations with different number of cells and mean polygon number 6. B: Box plots comparing the set of $\sqrt{SNR}$ values for configurations with different values of the mean polygon number and 127 cells. $D = 1000.0 \ \mu m^2/s$, $\alpha = 100.0 \ s^{-1}$. The $\star$ indicates statistically significant results (see Materials and methods for further details).
(PDF)

**S3 Fig. Mean and fluctuations of the LEGI readout variable as a function of the number of cells.** Mean (panels A-B) and variance (panels C-D) of the LEGI readout variable (Eqs (5) and (6)) in the weak-local (ISD) and weak (NNE) communication regimes, for the same configurations shown in Fig 5.
(PDF)

**S4 Fig. $\sqrt{SNR}$ as a function of the number of cells with different values of the mean polygon number.** Box plots comparing the set of $\sqrt{SNR}$ values for configurations with different number of cells, for different values of the mean polygon number (rows). The results are obtained combining 10 sets of cell configurations. Weak-local (ISD) communication: $D = 10.0$ $\mu m^2/s$, $\alpha = 1.0$ $s^{-1}$. Strong-local (ISD) communication: $D = 10.0$ $\mu m^2/s$, $\alpha = 100.0$ $s^{-1}$. Weak-global (ISD) communication: $D = 1000.0$ $\mu m^2/s$, $\alpha = 1.0$ $s^{-1}$. Strong-global (ISD) communication: $D = 1000.0$ $\mu m^2/s$, $\alpha = 100.0$ $s^{-1}$. Weak (NNE) communication: $\gamma_{NNE} = 0.1$ $s^{-1}$. Strong (NNE) communication: $\gamma_{NNE} = 10.0$ $s^{-1}$. The $\star$ indicates statistically significant results (see Materials and methods for further details).
(PDF)

**S5 Fig. Perimeter-area ratio for cell configurations with different mean polygon numbers.** Mean of the ratio between perimeter and square root of the area as a function of the mean polygon number for the same cell configurations shown in Fig 6F, in the Weak-global (ISD) communication regime: $D = 1000.0$ $\mu m^2/s$, $\alpha = 1.0$ $s^{-1}$.
(PDF)

**S6 Fig. $\sqrt{SNR}$ as a function of the mean polygon number for different number of cells.** Box plots comparing the set of $\sqrt{SNR}$ values for configurations with different values of the mean polygon number, for different number of cells (rows). The results are obtained combining 10 sets of cell configurations. Weak-local (ISD) communication: $D = 10.0$ $\mu m^2/s$, $\alpha = 1.0$ $s^{-1}$. Strong-local (ISD) communication: $D = 10.0$ $\mu m^2/s$, $\alpha = 100.0$ $s^{-1}$. Weak-global (ISD) communication: $D = 1000.0$ $\mu m^2/s$, $\alpha = 1.0$ $s^{-1}$. Strong-global (ISD) communication: $D = 1000.0$ $\mu m^2/s$, $\alpha = 100.0$ $s^{-1}$. Weak (NNE) communication: $\gamma_{NNE} = 0.1$ $s^{-1}$. Strong (NNE) communication: $\gamma_{NNE} = 10.0$ $s^{-1}$. The $\star$ indicates statistically significant results (see Materials and methods for further details).
(PDF)

**S7 Fig. Comparison between the ISD and NNE models for different number of cells.** Scatter plots of the mean of the square root SNR computed on the edge cells for 10 sets of configurations, for the ISD (*x*-axis) and NNE (*y*-axis) communication modes. In each plot we present the results for different values of the mean polygon number (MPN); results for different number of cells are shown in the rows. The dashed black line indicates the bisector. Error bars are given by the standard deviation. Weak-local communication: $D = 10.0$ $\mu m^2/s$, $\alpha = 1.0$ $s^{-1}$. Strong-local communication: $D = 10.0$ $\mu m^2/s$, $\alpha = 100.0$ $s^{-1}$. Weak-global communication: $D = 1000.0$ $\mu m^2/s$, $\alpha = 1.0$ $s^{-1}$. Strong-global communication: $D = 1000.0$ $\mu m^2/s$, $\alpha = 100.0$ $s^{-1}$.
(PDF)

**S1 Table. Statistical significance of the comparisons between $\sqrt{SNR}$ distributions for different number of cells.** We report the values of the *CLES*$-0.5$, which vary between $-0.5$ and $0.5$ (see Materials and methods for details), for the comparisons between the distributions of the SNR for different number of cells, shown in Fig 5 and S4 Fig. A comparison is considered statistically significant if the p-value from a Wilcoxon rank sum test is smaller than 0.05 and $|CLES - 0.5| > 0.1$. NS indicates that the comparison is not significant, according to this criterion.
(PDF)

**S2 Table. Minimum and maximum $\sqrt{SNR}$ for each value of the mean polygon number.** We report the minimum and maximum values of the average $\sqrt{SNR}$ for each value of the mean polygon number and in each regime of the parameters controlling the communication process. The average is computed over the sets of cell configurations for each value of the number of cells (cf. Fig 5, S2 and S4 Figs). Note that $MPN = 6$ includes also configurations with 169 and 217 cells.
(PDF)

**S3 Table. Statistical significance of the comparisons between $\sqrt{SNR}$ distributions for different mean polygon number.** We report the values of the $CLES-0.5$, which varies between $-0.5$ and $0.5$ (see Materials and methods for details), for the comparisons between the distributions of the SNR for different mean polygon number (MPN), shown in Fig 6. A comparison is considered statistically significant if the p-value from a Wilcoxon rank sum test is smaller than 0.05 and $|CLES - 0.5| > 0.1$. NS indicates that the comparison is not significant, according to this criterion. Communication regimes that are not shown in the table do not have statistically significant results. The last line shows the statistical significance between the extreme values of the mean polygon number interval used in the analysis.
(PDF)

**S4 Table. Minimum and maximum $\sqrt{SNR}$ for each value of the number of cells.** We report the minimum and maximum values of the average $\sqrt{SNR}$ for each value of the number of cells and in each regime of the parameters controlling the communication process. The average is computed over the sets of cell configurations for each value of the mean polygon number (cf. Fig 6, S2 and S6 Figs).
(PDF)

## Author Contributions

**Conceptualization:** Jonathan Fiorentino, Antonio Scialdone.

**Data curation:** Jonathan Fiorentino.

**Formal analysis:** Jonathan Fiorentino.

**Funding acquisition:** Antonio Scialdone.

**Investigation:** Jonathan Fiorentino.

**Methodology:** Jonathan Fiorentino, Antonio Scialdone.

**Project administration:** Antonio Scialdone.

**Resources:** Jonathan Fiorentino.

**Software:** Jonathan Fiorentino.

**Supervision:** Antonio Scialdone.

**Validation:** Jonathan Fiorentino, Antonio Scialdone.

**Visualization:** Jonathan Fiorentino.

**Writing – original draft:** Jonathan Fiorentino, Antonio Scialdone.

**Writing – review & editing:** Jonathan Fiorentino, Antonio Scialdone.

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
