## [Decision Letter · Decision Letter 0]

8 Dec 2021

Dear Dr Scialdone,

Thank you very much for submitting your manuscript "The role of cell geometry and cell-cell communication in gradient sensing." for consideration at PLOS Computational Biology.

As with all papers reviewed by the journal, your manuscript was reviewed by members of the editorial board and by several independent reviewers. In light of the reviews (below this email), we would like to invite the resubmission of a significantly-revised version that takes into account the reviewers' comments.

We cannot make any decision about publication until we have seen the revised manuscript and your response to the reviewers' comments. Your revised manuscript is also likely to be sent to reviewers for further evaluation.

Sincerely,

Philip K Maini

Associate Editor

PLOS Computational Biology

Jason Haugh

Deputy Editor

PLOS Computational Biology

Reviewer's Responses to Questions

**Comments to the Authors:**

Reviewer #1: In this paper the authors address the question of understanding how cell geometry and cell-cell communication can affect gradient sensing. They use as starting point the local-excitation global-inhibition model (LEGI) and generalise it to investigate the role played by cell number, geometry and long- and short- cell-cell communication in efficiently measuring concentration gradients. To this aim they define two different modes of cell-cell communication, i.e., the nearest neighbour exchange (NNE) where a global reporter allows communication only between nearest neighbour cells, and the Intercellular Space Diffusion (ISD), where the global reporter can be secreted by the cells, diffuse and internalised by other cells. This further ingredient, besides being an elegant way of modelling cell-cell communication, also allows them to span a wide range of interactions according to the diffusion length of the global reporter. Starting from this model and these different interactions schemes, they study the Signal-to-Noise Ratio to quantify the precision of gradient sensing. In summary, their results show that i) according to the type of communication, the precision of gradient sensing can depend on the number of cells. In particular, when the communication is weak and local, smaller systems perform better, whereas when the communication is global and/or strong, larger systems perform better. ii) By varying the tissue geometry, i.e., number of edges per polygon for example, in the regime of fast diffusion in the ISD model, more disordered configurations lead to more precise gradient sensing, whereas ordered structures are beneficial in the regime of nearest neighbours communication.

I find the paper very interesting, well done and well written. I appreciated a lot the extensive explanations about their modelling approach, analysis and obtained results and the literature is well represented. I only have a few minor points which I'd like the authors might clarify and I believe could improve the quality of the manuscript.

i) In the paper, I have noticed that the authors never explicitly mention the variety of differently diffusing morphogens which actually play the same role as their global reporter in different species (see for example Delta-Notch in Drosophila, Nodal-Lefty in zebrafish, Wnt in a number of different species). I think it would be important to discuss this and to also relate their results to some of the morphogens whose diffusion constants have been estimated.

ii) my main concern lies in the comparison of the Signal-to-Noise Ratios between the ISD and NNE modes of communication in Fig. 5 and 6. In Fig 5 panels A and B it does seem that there's an optimum around 37 cells, which the authors explain in the text. Yet, I think, if it's possible, it'd be better to extend the simulations further beyond 127 cells to clearly see whether there's a real decrease in the SNR. Same for Fig 5E where instead it seems that the behaviour of SNR tends to saturation. In Fig 6, instead, in particular panel B and E, I believe it would be better to extend the mean polygon number to a wider range (e.g., between 3 and 8?) as well as make sure that the parameters used in the model as optimal in order to observe the highest variations in the plots. At the moment, in fact, the fold decrease/increase in panels E and B is quite small to make actual claims in my opinion (or is there a point which I am missing?). Also, I would honestly move Fig S3-S4 directly in the Main Text as in my opinion they summarise most of the results and are amazingly clear. To better highlight the results, I would also use the same scale across all the different panels.

iii) Reference 38 appears to miss the authors.

Reviewer #2: In this manuscript, the Authors propose and computationally characterize a model of collective gradient sensing for epithelial cell monolayers, where distant cells communicate with each other by secreting a molecular reporter which can diffuse along cell-cell contacts.

The model belongs to the class of ‘local excitation, global inhibition’ (LEGI) models, and generalizes a similar model previously introduced in 2016 by Levchenko and collaborators.

The simple idea behind LEGI models is that a cell, or, in these more recent proposals, a group of cells, would locally produce a molecular factor proportional to a sensed external gradient, while a second, rapidly diffusing molecular factor would ‘compute’ an average of the external signal, to be subtracted to the signal itself, so that the resulting (positive and negative) differences could then be amplified to give opposing responses.

In the approach proposed here, the role of the global inhibitor is played by a hypothetical molecular reporter which is secreted by the epithelial cells along cell-cell contacts, diffuses along cell-cell contact lines, and is later internalized either by nearest-neighbor cells, or by more distant cells.

My greater perplexity about this work regards this specific hypothesis, as I don't know of evidence supporting the idea that a molecular reporter would travel specifically in the quasi one-dimensional region of cell-cell contacts. In their Introduction, the Authors name as possible candidates small molecules such as ATP, calcium ions and NO; soluble ligands that bind EGFR; and extracellular vesicles. However, small molecules are not likely to be confined to the region of cell-cell contacts. Perhaps, one may imagine that tight junctions could confine the diffusion of larger molecular reporters to the quasi one-dimensional cell-cell contact region. Or, that confinement could be implemented by rapid association and dissociation to/from receptors specifically concentrated along cell-cell contacts. But in any case, this main conceptual difficulty should be discussed. Either the Authors know about, or at least suspect the existence of possible biological realizations of their abstract model, and in that case, the paper should be integrated with a discussion of these examples. Or their model is purely speculative, and in that case the speculative character of the hypotheses should be more clearly stated.

Secondly, other ways (for instance, mechanical) of introducing long-range intercellular communication for collective gradient sensing have been proposed: a discussion of these alternatives would therefore be in order.

Apart from these main difficulties, the paper is well conceived and well written. The model is clearly exposed and thoroughly characterized, the computational and statistical methods used are sound. In the framework of the model, it is shown that long-range communication would be beneficial to collective gradient sensing, especially when low levels of molecular reporters are secreted. Additionally, ‘disordered’ epithelial monolayers are shown to be able to sense gradients with a marginally higher signal-to-noise ratio, if compared to perfectly ordered (purely hexagonal) monolayers.

In conclusion, I do not think that in its present version this manuscript has the level of significance for the discipline which is required for publication in PLOS Computational Biology. That would change in my opinion if the Authors were able to convincingly make the case for the plausibility of their model.

Minor observations:

- Sect. 3.3: it is shown that in the ‘weak-global’ regime, the SNR is higher in configurations with lower mean polygon number; this is interesting, and it would be nice to have an intuitive understanding of this effect; perhaps it is related to the fact that if the area is kept fixed, the perimeter of polygons with a smaller number of edges is smaller?

- Fig. 6.F: it would be useful to explain the rationale for dividing the distances by the average edge length.

- Eq. 3: there is a wrong index (n in the place of k); actually, it would be more clear to have everywhere (in Eqs. 3,4 and in the text) the couple of indices k,k'; or, alternatively, n,n'.

- Eq. 12: a differential of tau seems to be missing in the second integral; also, the use of the letter d for both the distance and the differential is confusing; perhaps it could be convenient to use s instead; it could also be useful to have a reference (for instance, Redner 2001) for the first-passage time formula used.

- Fig. 3, caption: ‘edges represent . . . ’; perhaps: ‘edge thickness represent . . . ’? it is not clear how exactly the different weights are represented in the weighted graph.

- Fig. 1, caption: there is a mention of ‘edge cells’, but the notion of ‘edge cells’ has not been discussed previously.

- The year is missing in Ref. 15.

Reviewer #3: Re: Review of "The role of cell geometry and cell-cell communication in gradient sensing" by Jonathan Fiorentino and Antonio Scialdone.

The paper investigates how groups of cells collectively sensing shallow gradients of external systems. Cellular communication is modelled using a local-excitation global-inhibition (LEGI) model. A LEGI model has two components: (1) a local reported confined to a cell representing a local measurement of the concentration, and (2) global reported molecules produced in response to the external signal, which is exchanged with neighbouring cells. Previous work includes using LEGI models to study collective sensing of external gradients in one-dimensional and very simple two-dimensional geometries. This paper extends these results to more complicated (and realistic) two-dimensional geometries. The authors investigate how the signal-to-noise ratio (SNR) depends on the number of cells, geometry, local vs. global communication mechanism, and strength of nearest neighbour communication.

In more detail, two different models of cell-cell communication are considered: (1) nearest neighbour exchange of the global reporter (short-ranged), and (2) intercellular space diffusion (ISD) of the global reported on the cell-cell interfaces (long-ranged). For the ISD model the nonlocal exchange rates between cells are estimated by computing 5000 trajectories of diffusing global reported molecule (for each geometry).

Result Summary

The authors find that:

- Smaller systems (with respect to cell number) tend to have higher SNR when cell communication is weak or local.

- Larger systems have higher SNR when cell communication is global (with ISD) and/or strong.

- The NNE / ISD have similar SNR in the strong-local regime i.e. when the reporter is mostly local and efficiently exchanged between cells.

- Finally the authors investigated the effect of geometry on the SNR. Interestingly, for the ISD model more irregular geometries have higher SNR.

Conclusion

The results of this study are particularly significant to epithelial tissues which feature dynamically changing geometries, since it is currently unknown how such geometric arrangement affect

collective gradient sensing. The insights of this study are of high interest to modellers of epithelial tissues (e.g. cellular Potts models, vertex models etc.). In particular, those modelers

considering processes depending on external gradient sensing (e.g. development). Finally the study is interesting, well written and well presented. The code for studies' results appear to be completely available on github. I recommend this study for publication.

Specific Comments:

On line 103 you define the signal to noise ratio as your primary objective.

- It might be beneficial for the reader to define ``good'' or ``bad'' values of the SNR, so that a reader has some reference points in mind when encountering the models results.

- I'm confused by the line "To compute it, we use the formula obtained in [18]". What is it here? Is it the SNR? Because isn't the SNR computed using equations (5) and (6)?

Minor comments / questions:

Line 22: What do you mean with physical limits?

**Have the authors made all data and (if applicable) computational code underlying the findings in their manuscript fully available?**

Reviewer #1: Yes

Reviewer #2: Yes

Reviewer #3: Yes

PLOS authors have the option to publish the peer review history of their article (what does this mean?). If published, this will include your full peer review and any attached files.

Reviewer #1: No

Reviewer #2: No

Reviewer #3: No
---

## [Decision Letter · Decision Letter 1]

17 Feb 2022

Dear Dr Scialdone,

We are pleased to inform you that your manuscript 'The role of cell geometry and cell-cell communication in gradient sensing.' has been provisionally accepted for publication in PLOS Computational Biology.

Before your manuscript can be formally accepted you will need to complete some formatting changes, which you will receive in a follow up email. A member of our team will be in touch with a set of requests. As you prepare your final manuscript, please also consider incorporating the suggestions by Reviewer #2.

Best regards,

Philip K Maini

Associate Editor

PLOS Computational Biology

Jason Haugh

Deputy Editor

PLOS Computational Biology

Reviewer's Responses to Questions

**Comments to the Authors:**

Reviewer #1: The authors have addressed all my major concerns and thus I recommend the paper for publications.

Reviewer #2: The Authors satisfactorily responded to all of my concerns. I support publication of their manuscript in PLOS Comp. Biol., assuming that a couple of remaining observations are also taken into account.

Residual observations about the present, revised version:

- the acronym LIS is not spelled out the first time it appears in the text (line 245)

- the reason for the introduction of the concept of ‘lateral intercellular space’ in line 245 is not given in the text; one would expect some discussion of the kind: “in the present model, diffusion was restricted to the onedimensional lateral intercellular space (LIS)’; however, this condition is not as restrictive as it could appear; in the SI we also consider . . . and show that . . . ”

**Have the authors made all data and (if applicable) computational code underlying the findings in their manuscript fully available?**

Reviewer #1: Yes

Reviewer #2: Yes

PLOS authors have the option to publish the peer review history of their article (what does this mean?). If published, this will include your full peer review and any attached files.

Reviewer #1: No

Reviewer #2: No

---

## [Editor Report · Acceptance letter]

8 Mar 2022

PCOMPBIOL-D-21-01851R1 

The role of cell geometry and cell-cell communication in gradient sensing.

Dear Dr Scialdone,

I am pleased to inform you that your manuscript has been formally accepted for publication in PLOS Computational Biology. Your manuscript is now with our production department and you will be notified of the publication date in due course.

With kind regards,

Zita Barta
